# Mortality and Medical Complications of Subtrochanteric Fracture Fixation

**DOI:** 10.3390/jcm10030540

**Published:** 2021-02-02

**Authors:** Michalis Panteli, Marilena P. Giannoudi, Christopher J. Lodge, Robert M. West, Ippokratis Pountos, Peter V. Giannoudis

**Affiliations:** 1Academic Department of Trauma & Orthopaedics, School of Medicine, University of Leeds, Leeds LS1 3EX, UK; mgiannoudi@aol.com (M.P.G.); I.Pountos@leeds.ac.uk (I.P.); pgiannoudi@aol.com (P.V.G.); 2Leeds General Infirmary, Great George Street, Leeds LS1 3EX, UK; christopherlodge@nhs.net; 3Leeds Institute of Health Sciences, University of Leeds, Leeds LS2 9JT, UK; R.M.West@leeds.ac.uk; 4NIHR Leeds Biomedical Research Unit, Chapel Allerton Hospital, Leeds LS7 4SA, UK

**Keywords:** hip fractures, subtrochanteric, complications, mortality

## Abstract

The aim of this study was to define the incidence and investigate the associations with mortality and medical complications, in patients presenting with subtrochanteric femoral fractures subsequently treated with an intramedullary nail, with a special reference to advancement of age. **Materials and Methods:** A retrospective review, covering an 8-year period, of all patients admitted to a Level 1 Trauma Centre with the diagnosis of subtrochanteric fractures was conducted. Normality was assessed for the data variables to determine the further use of parametric or non-parametric tests. Logistic regression analysis was then performed to identify the most important associations for each event. A *p*-value < 0.05 was considered significant. **Results:** A total of 519 patients were included in our study (age at time of injury: 73.26 ± 19.47 years; 318 female). The average length of hospital stay was 21.4 ± 19.45 days. Mortality was 5.4% and 17.3% for 30 days and one year, respectively. Risk factors for one-year mortality included: Low albumin on admission (Odds ratio (OR) 4.82; 95% Confidence interval (95%CI) 2.08–11.19), dementia (OR 3.99; 95%CI 2.27–7.01), presence of pneumonia during hospital stay (OR 3.18; 95%CI 1.76–5.77) and Charlson comorbidity score (CCS) > 6 (OR 2.94; 95%CI 1.62–5.35). Regarding the medical complications following the operative management of subtrochanteric fractures, the overall incidence of hospital acquired pneumonia (HAP) was 18.3%. Patients with increasing CCS (CCS 6–8: OR 1.69; 95%CI 1.00–2.84/CCS > 8: OR 2.02; 95%CI 1.03–3.95), presence of asthma/chronic obstructive pulmonary disease (COPD) (OR 2.29; 95%CI 1.37–3.82), intensive care unit (ICU)/high dependency unit (HDU) stay (OR 3.25; 95%CI 1.77–5.96) and a length of stay of more than 21 days (OR 8.82; 95%CI 1.18–65.80) were at increased risk of this outcome. The incidence of post-operative delirium was found to be 10.2%. This was associated with pre-existing dementia (OR 4.03; 95%CI 0.34–4.16), urinary tract infection (UTI) (OR 3.85; 95%CI 1.96–7.56), need for an increased level of care (OR 3.16; 95%CI 1.38–7.25), pneumonia (OR 2.29; 95%CI 1.14–4.62) and post-operative deterioration of renal function (OR 2.21; 95%CI 1.18–4.15). The incidence of venous thromboembolism (VTE) was 3.7% (pulmonary embolism (PE): 8 patients; deep venous thrombosis (DVT): 11 patients), whilst the incidence of myocardial infarction (MI)/cerebrovascular accidents (CVA) was 4.0%. No evidence of the so called “weekend effect” was identified on both morbidity and mortality. Regression analysis of these complications did not reveal any significant associations. **Conclusions:** Our study has opened the field for the investigation of medical complications within the subtrochanteric fracture population. Early identification of the associations of these complications could help prognostication for those who are at risk of a poor outcome. Furthermore, these could be potential “warning shots” for clinicians to act early to manage and in some cases prevent these devastating complications that could potentially lead to an increased risk of mortality.

## 1. Introduction

Internationally, countries are seeing a proportional increase in their elderly population. In the United Kingdom (UK), the 2016 review by the Office of National Statistics identified that 18% of the UK population was over the age of 65 [1]. With improved healthcare and the resultant increase in life expectancy, this is predicted to continue to grow with estimates for 2041 reaching 25% [1]. This increase in ageing population will increase both health and social care spending accounting for an additional 2% gross domestic product (GDP) by 2060 [2,3].

Within the elderly population, hip fractures represent the most common injury requiring surgical intervention and are associated with significant morbidity and mortality. More than 70,000 patients suffer from a hip fracture every year in the United Kingdom alone, with the total cost of their care being higher than £2 billion [4]. Subtrochanteric fractures, in particular, are unique in that they lack a consensus regarding their definition. The most commonly accepted definition is that they include fractures that occur within 5 cm distal to the lesser trochanter [5]. Subtrochanteric fractures pose additional challenges in their management, because of the unique anatomical and biomechanical features of the subtrochanteric region [6], their “moderate” vascularity [7], in addition to the reduction difficulties associated with the strong deforming forces acting on the proximal femur [8]. The mainstay of their management is intramedullary nailing (>90%) [9], due to the biomechanical advantages offered from this method of fixation compared to others [5]. Most notably, following an otherwise “successful” treatment, the incidence of mortality remains high and has been reported at 10% for the first month and 30% for the first year thereafter [9]. Medical complications including delirium, systemic infections, venous thromboembolism (VTE), myocardial infarctions (MI) and cerebrovascular accidents (CVA) are also common [9].

The primary aim of this study is to assess the incidence of mortality in patients presenting with subtrochanteric femoral fractures and to evaluate any potential associations with mortality at one-year post injury. Secondly, we aim to obtain the incidence of complications associated with their surgical management and in-hospital stay (concentrating on VTE and cardiac events, systemic infections and post-operative delirium), and to determine whether a “weekend effect” actually exists in this cohort of patients.

## 2. Patients and Methods

Following institutional board approval (LTH#2591), we performed a retrospective review of all patients admitted to our unit (Level 1 Trauma Centre) between 1 January 2009 and 31 December 2016 (8-year period) with the diagnosis of a subtrochanteric fracture. These patients were identified following review of hospital admission records and theatre lists. All patients with the aforementioned injury, bar the exclusions listed below, were included in this study. Skeletally immature patients, patients having their operation in other institutions, patients transferred to other institutions following their operations and pathological fractures were excluded from further analysis. In the case of bilateral fractures (same or different episodes), only the first episode was considered for the analysis.

Written and electronic medical records, daily trauma lists, theatre records and electronic platforms of radiographs, pathology and microbiology results were used for data collection. This included; patients’ demographics, injury characteristics including mechanism of injury and associated injuries, past medical history, (including comorbidities to enable classification as per Charlson comorbidity score (CCS)), drug history and social history, fracture characteristics, primary operation details, details of any additional procedures required until union is achieved, length of hospital stay, mortality, cause of death, time to radiographic/clinical union, morbidity related to admission (including implant, fracture and medical related complications) and pathology results (including blood tests, microbiology and histology reports).

Follow up duration was as per our departmental protocol, with varying durations of patient follow up dependant on patient age/function. Young or active patients were followed up until radiological/clinical union has occurred. Elderly patients with impairments of mobility or mental function were only followed up in the case of an arising complication, typically via a re-referral from their general practitioner (GP). Equally, any patients discharged following union of their subtrochanteric fracture who subsequently develop a complication could re-enter the service via general practitioner referral or direct re-attendance to the accident and emergency department. Patient death was verified through hospital electronic records, along with GP records to ensure validity of data.

### Statistical Analysis

All collected data was inserted into an electronic database. Statistical analysis was undertaken using the computing environment R (R version 3.6.0; R Core Team (2019). R: A language and environment for statistical computing. R Foundation for Statistical Computing, Vienna, Austria. URL: https://www.R-project.org/). Basic demographic data were presented as count (percentage) or as mean ± SD. Parametric data were analysed using a Welch unpaired independent *t*-test, whilst count data were analysed using a Pearson’s chi square test. A *p*-value of <0.05 was considered significant. Mortality findings were graphically presented using Kaplan–Meier survival curves.

For the identification of the significant variables associated with each condition, the statistical analysis started with unadjusted analysis. From all the variables investigated, only those with a *p* < 0.05 significance were included for the revised adjusted model, using a logistic regression analysis. For the final model, only those variables having a *p*-value of less than 0.05 were retained and odds ratio (OR) were reported.

## 3. Results

### 3.1. Patient Demographics and Mechanism of Injury

A total of 519 patients (age at time of injury: 73.26 ± 19.47 years; 318 female) were identified and formed the basis of this study. The majority of patients within this study were within the elderly cohort, with 391 patients (75.3%) being ≥65 years old.

The incidence of these injuries within the male population had a bimodal distribution with the first peak at ages 30–40, and the second at ages 80–90, which included a greater proportion of patients. Conversely, with female patients, incidence increased with age. Comorbidities included a previous history of malignancy in 103 patients (lung: 33 patients, breast: 26 patients and bowel: 18 patients), dementia in 120 patients and diabetes in 69 patients. A total of 103 patients were active smokers. The majority of the patients were independently mobile (271 patients—52.2%) and only a small proportion were residential or nursing home residents (34 and 5 patients, respectively), whilst another 46 patients had regular carers.

The majority of fractures (82.9%) were sustained as a result of “low impact” injury, which is in keeping with elderly trauma. The most common mechanism of injury was “fall from standing height” (62.2%), followed by “unwitnessed fall” (12.9%). Higher impact mechanisms of injury including “road traffic accidents” (RTAs) were recorded at 9.1%. Given the assessment of age-related impact on this patient group we analysed the prevalence of “bone protection” intake. Less than one quarter of all patients were on bisphosphonates prior to admission (87 patients—16.7%), and just under one third were on calcium or vitamin D replacement (149 patients—28.7%).

### 3.2. Hospital Stay

The average length of hospital stay was 21.4 days (median: 18 days; SD: 19.45 days). Unsurprisingly the length of stay increased with age. In the younger population increased length of stay was associated with polytrauma (Injury Severity Score (ISS) > 16) and a higher incidence of intensive care unit (ICU)/high dependency unit (HDU) stay. Of the patients with prolonged hospital stay (over 28 days), 16 patients were aged less than 65 years of age, whilst 53 patients were aged greater than 85 years of age.

### 3.3. Mortality

Regardless of age, 30-day mortality was 5.4%, compared to 17.3% for one-year mortality. In patients older than 65 years old (391 patients), 30-day mortality increased to 7.6% and one-year mortality to 25.6%. One-year mortality (regardless of age) was higher in female patients even though this was not statistically significant (*p* = 0.575), but it was significantly higher with an increasing age (*p* < 0.001). Kaplan–Meier survival curves for overall mortality on the other hand demonstrated a statistically significant higher mortality risk associated with female gender and age (*p* < 0.001 and *p* = 0.021 respectively) (Figure 1). Average time to death from injury was 25.7 months (median: 21.4 months; SD: 23.1 months). Thirty-day mortality was positively correlated with age, with no patient younger than the age of 65 years dying within 30 days and no patient younger than the age of 40 years dying within the first-year post injury. The comparison of the different parameters according to one-year mortality is presented in Appendix A, whilst the significant unadjusted associations are presented in Table 1. Interestingly, within our cohort timing of operative intervention (i.e., operation within 48 h from admission) had no impact on mortality (*p* = 0.871).

Following an adjusted regression analysis, several factors were found to be associated with one-year mortality (Table 2): Low albumin on admission (OR 4.82; 95% confidence interval (95%CI) 2.08–11.19), dementia (OR 3.99; 95%CI 2.27–7.01), presence of pneumonia during hospital stay (OR 3.18; 95%CI 1.76–5.77) and CCS > 6 (OR 2.94; 95%CI 1.62–5.35).

### 3.4. Venous Thromboembolism

A total of 19 venous thromboembolism events (incidence 3.7%), including pulmonary embolism (PE) in 8 patients and deep vein thrombosis (DVT) in 11 patients, were documented within six months from the index procedure. Unadjusted associations to occurrence of VTE were; body mass index (BMI) > 30 (*p* < 0.001), post-operative transfusion (*p* < 0.001) and wound infections (*p* < 0.001). Nonetheless, there was no association of VTE with mortality, hospital stay, need for ICU/HDU admission or surgical complications and no factor remained significant after adjusting for cofounders.

### 3.5. Myocardial Infarction/Cerebrovascular Accidents

A total of 21 patients, incidence 4.0% suffered from myocardial infarction or cerebrovascular accidents within six months of the index procedure. Patients younger than the age of 65 were less likely to have MI/CVA, compared to those older than 75 years old (*p* = 0.016 and *p* = 0.009, respectively). There was no association with smoking or alcohol habits prior to admission. Surprisingly, a longer surgical time was less predictive of MI/CVA (*p* = 0.048). The American Association of Anaesthesiologists (ASA) grade was however implicated as an associated factor (*p* = 0.003), with higher incidence being seen with higher ASA grades. This was also reflected on CCS (*p* = 0.001). Whilst MI/CVA did not predict ICU/HDU stay, it did predict 30-day and one-year mortality (*p* < 0.001 and *p* = 0.004, respectively). Nevertheless, when we attempted to build a regression analysis model, we failed to identify any factors that remained significant.

### 3.6. Hospital Acquired Pneumonia

A total of 95 (incidence 18.3%) patients suffered from hospital acquired pneumonia (HAP). Age < 65 years old (*p* < 0.002) and age > 75 years of age. (*p* = 0.022) were, respectively, negatively and positively correlated to development of HAP. Furthermore, the patients’ ASA score prior to surgery was also associated with HAP, as was a higher CCS (*p* = 0.001), presence of diabetes (*p* = 0.022) and asthma/chronic obstructive pulmonary disease (COPD) (*p* < 0.001). Interestingly, smoking was not associated with a risk of developing HAP. HAP was also a predictor of HDU/ICU stay (*p* < 0.001), as well as length of hospital stay, and both 30-day and one-year mortality (*p* < 0.001). Time to surgery, operation characteristics and surgical complications were not associated with an increased risk of developing HAP.

Following an adjusted regression analysis, development of HAP was associated with: an increasing CCS (CCS 6–8: OR 1.69, 95%CI 1.00–2.84; CCS > 8: OR 2.02, 95%CI 1.03–3.95), presence of asthma/COPD (OR 2.29; 95%CI 1.37–3.82), ICU/HDU stay (OR 3.25; 95%CI 1.77–5.96) and a length of stay of more than 21 days (OR 8.82; 95%CI 1.18–65.80) (Table 3).

### 3.7. Post-Operative Delirium

Post-operative delirium was diagnosed in 53 cases within our cohort (incidence 10.2%). Patients aged < 65 years old had a lower incidence of post-operative delirium (*p* < 0.001), in contrast patients older than 75 years of age (*p* = 0.001). There was no association with gender, but the incidence was higher in isolated injuries compared to patients sustaining more than one injury (*p* = 0.005). Once more, increasing ASA grade and CCS were implicated with a higher risk (*p* = 0.023 and *p* < 0.001, respectively), as was presence of dementia (*p* < 0.001). Additionally, poor pre-operative mobility (*p* = 0.007) and patients sustaining frequent falls (*p* = 0.014) were also linked to an increased risk. Moreover, delirium was associated with HAP and urinary tract infections (*p* = 0.003 and *p* < 0.001, respectively), as well as pre- and post-operative chronic kidney disease (CKD) (*p* = 0.017 and *p* < 0.001) and need for post-operative transfusion (*p* = 0.021). Whilst there was no association between delirium and mortality, delirium correlated with HDU/ICU stay and increased length of stay in hospital (*p* = 0.015 and *p* < 0.001, respectively).

Following an adjusted regression analysis, dementia (OR 4.03; 95%CI 0.34–4.16), urinary tract infection (UTI) (OR 3.85; 95%CI 1.96–7.56), need for an increased level of care (OR 3.16; 95%CI 1.38–7.25), pneumonia (OR 2.29; 95%CI 1.14–4.62), and post-operative deteriorating renal function (OR 2.21; 95%CI 1.18–4.15) were strongly associated with development of post-operative delirium (Table 4).

### 3.8. Weekend Effect

Most patients within our cohort were admitted on a weekday (350 patients—67.4%). There was no significant difference in terms of patients’ demographics admitted each day of the week. High-energy injuries were more prevalent over the weekends (19.0% versus 14.7%), as was patients with open fractures (*p* = 0.071). Even though ASA was similar in the two groups, there was a trend for a higher CCS over weekdays (*p* = 0.086), but this was not significant. Interestingly, there was no difference to time taken from admission to the operating room. More specifically, weekday admissions had operations performed in less than 48 h in 78.8% of patients, compared to 79.9% of cases during weekends. In keeping with this there was no difference in the level of the first surgeon performing the surgery, the presence of a consultant in theatres, as well as the surgical and anaesthetic times. Finally, there was no significant difference in the incidence of complications including HAP, superficial and deep wound infections, VTE or mortality (*p* > 0.100).

## 4. Discussion

In this study we attempted to investigate the outcomes of operative management of subtrochanteric fractures treated with an intramedullary nail, with specific reference to the associated mortality and medical complications within the elderly population. As per our primary aim, the difference in early mortality between the different age groups was most notable. There was no documented death in patients aged 65 or under within the first 30 days, and there was no reported mortality at one year in patients younger than the age of 40 years old. One feasible rationale is that the younger cohort either sustain terminal trauma at the scene (and therefore are not included in our study) or have such high physiological reserve that they are able to overcome the first 30 days post injury. In contrast, we found increasing mortality levels positively associated with age which likely represents the ever-depleting physiological reserve within patients sustaining fragility fractures, a finding previously reported in the literature [10].

The 30-day mortality rate in our entire patient cohort was tallied at 5.4%, with one-year mortality rising to 17.3%. When patients younger than the age of 65 years were excluded, mortality increased to 7.6% and 25.6%, respectively. This is marginally higher than the national all-cause hip fracture mortality data of 30-day mortality being quoted at 6.9% (cf. 6.7% in 2016) [9]. Our mortality results are also similar to a Swedish registry study of both subtrochanteric and trochanteric hip fractures, where 30-day mortality was reported 7.7% and one-year mortality at 25.9% [11]. With regards to gender, in contrast to findings of other studies, we were unable to identify any increased mortality in male patients [10,11]. On the contrary, mortality was found to be slightly higher in female patients in the course of 10 year follow up, this however was not of statistical significance.

The presence of comorbidities (CCS > 6) significantly increased the risk of mortality. Comorbidities in general, modify the course of a disease such as in the case of a hip fracture. Several authors have reported an increasing rate of mortality with an increase in comorbidities [12,13,14,15]. More specifically, Lunde et al. in an epidemiological study reported a 15% increase in deaths during the first year in women with a CCS score of ≥3 [12], findings comparable to those of Jürisson et al. [15]. Dementia in particular, was found to be an independent predictor of mortality in our cohort, confirming previously reported findings [12,16,17]. Additionally, the presence of pneumonia significantly increased mortality, which was again in line with previous findings [18,19].

Presence of a low serum albumin level on admission also significantly increased the mortality risk, a finding previously reported by other authors, quoting a 2.5-time increase in mortality in patients with hypoalbuminaemia [20,21,22,23]. Serum albumin is a marker of “nutritional state” and low levels are associated with low muscle mass, strength and function. However, reduction in serum albumin could also reflect the existing comorbidities, chronic illness and concurrent infection [22]. Nutritional supplementation is warranted in order to reduce both the risk of developing complications, as well as the risk of mortality [24].

Venous thromboembolism is a well-reported complication for any patient following surgical intervention. Within the hip fracture population, work has suggested that the risk of VTE is in keeping with the degree of comorbidity [25]. The literature also suggests that in order to avoid VTE, early surgery (within 48 h) and pharmacological (or in combination with mechanical) thromboprophylaxis is vital [26,27]. The incidence of asymptomatic VTE in hip fracture patients without thromboprophylaxis is between 42%–50% [28]. Within our cohort, 4% of patients developed VTE. The presence of VTE had no clinical impact on patients requiring higher levels of care (admission to ICU/HDU), nor did it impact their length of hospital stay. We suggest that our pathway of early mobilisation and prompt review by an orthogeriatric contributed to these good outcomes.

Within this study, MI/CVA events relating to the index admission was 4.0%. This included, pre-, peri- and post-operative events (up to 6 months post–surgery). This is slightly higher than a large study looking at over two million hip fracture operations, which quotes a total number of adverse cardiovascular events to be at 3.3% [29]. However, this study only investigated peri-operative events and may have underestimated the risk because of “under-coding” of these conditions. In line with our findings, MI/CVA has been reported as a predictor of increased mortality [12,28,30]. Our data also suggests that increasing ASA grade is associated with increasing risk of cardiovascular adverse events. This has previously been reported in hip fracture research though not within the subtrochanteric fracture population [31]. Factors to consider in the interpretation of this, is that the ASA considers comorbidities only, not age nor sex which can all contribute to the overall outcome of a patient’s treatment.

Approximately one in five of the patients in our cohort (18.3%) developed HAP during their admission. This is significantly higher than the figures quoted in the literature of 4%–7% [28,32,33,34], but differences in the methodology of the studies may account for this. Furthermore, there has been no study, to date, looking directly at subtrochanteric fractures and incidence of HAP as oppose to all cause hip fractures. This could therefore, potentially, represent a higher risk with this specific type of fracture, especially with the highest incidence of weightbearing protection and reduced mobility following the operative management of such complex injuries. In keeping with previous reports, our study displayed that the incidence of HAP increased with age [33]. Risk factors for HAP, previously identified, include; male sex, older age (especially ≥90 years), low body mass index and chronic lung disease. These comorbidities have also been associated with increased risk of complications, consequent escalation of treatment and increased length of stay [18,32,35,36]. Our findings support this, with increasing comorbidities and chronic lung disease being the main risk factors for HAP, whilst this was also associated with an increased length of stay and risk of a longer HDU/ICU stay.

Delirium has been quoted to have an accumulated incidence of 24%–33% amongst the elderly population with hip fractures [37], compared to 10.2% in our series. McCusker et al. found delirium to be an independent predictor of one-year mortality, with its effect being more serious for patients without pre-existing dementia [38]. Furthermore, it has been suggested that the timing of delirium can affect mortality, i.e., immediate post-operative delirium (within 24 h) is implicated with higher mortality, compared to delayed (>24 h post-operatively) [38]. This is controversial within the literature however, with some suggesting delirium has no impact on the survival of patients with hip fractures [39]. In a study by Harris et al., pre-existing dementia was the factor with the greatest risk for developing post-operative delirium [40]. Our regression analysis supports this, suggesting that delirium was most likely in patients with underlying dementia (most important predictive factor), infections (UTI and pneumonia) and a deteriorating renal function. Therefore, a multi-disciplinary approach with early assessment by geriatricians and monitoring for delirium, prevention and early treatment of infection, correction of biochemical dehydration with adequate fluids and medicines optimisation could potentially reduce this complication.

In regard to the so called “weekend effect”, the literature contains overwhelming evidence to suggest that there is in fact no “weekend effect” for the treatment of patient with hip fractures [41,42,43]. Whilst there has been no specific study to investigate this within the subtrochanteric fracture population, our data correlates with that of the literature.

To our knowledge this is the first study to investigate medical complications in patients specifically presenting with subtrochanteric fractures. These injuries have been historically suggested to be more “malignant”. Our data can be used as the foundation for further work within this area. Limitations include the retrospective nature of the study, where data collection was limited to that which had already been documented by the clinical team. Compared to registry studies, our cohort was relatively small and hence this could also skew our results. However, we believe that given the prudent investigation of all patient admission documents, our data has adequately represented our cohort. Finally, due to the lack of research within this field, it is difficult to fully compare our findings with that of the literature given the potential for different risk factors and effects of subtrochanteric fractures to those of hip fractures in general.

In conclusion, our study has explored the relationship between aging and mortality in the subtrochanteric fracture population. It has also opened the field for the investigation of medical complications within the subtrochanteric fracture population. The bi-modal age distribution in patients suffering from these fractures reveals once again, that the elderly are at higher risk of increased morbidity and mortality. Identification and modification of the risk factors associated with complications and mortality, may help improve outcome. This potentially could be achieved through an introduction of stricter best-practice guidelines by the multidisciplinary team to promptly optimise the care of this patient cohort and to reduce the associated high healthcare costs.

## Figures and Tables

**Figure 1 jcm-10-00540-f001:**
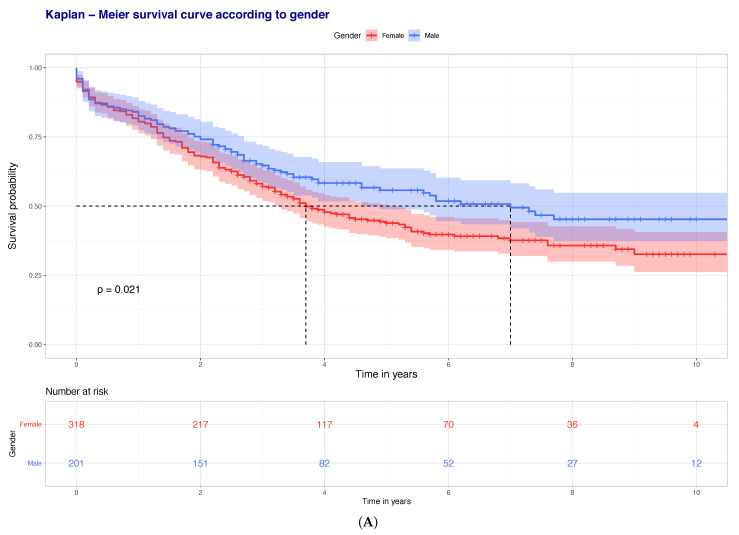
Kaplan–Meier survival curves (one-year mortality) according to (**A**) gender and (**B**) age of patients.

**Table 1 jcm-10-00540-t001:** Statistically significant unadjusted associations with one-year mortality.

**Demographics**	**Unadjusted OR** **(95% CI)**	***p*-value**
Age ≥ 75	5.87 (3.04–11.35)	<0.001
**Injury Characteristics**	**Unadjusted OR** **(95% CI)**	***p*-value**
ISS > 16	0.13 (0.02–0.97)	0.046
**Patient Co-morbidities**	**Unadjusted OR** **(95% CI)**	***p*-value**
Dementia	5.67 (3.49–9.21)	<0.001
Pre-operative CKD	2.32 (1.44–3.72)	<0.001
Post-operative CKD	2.70 (1.67–4.35)	<0.001
Hypoalbuminaemia	6.23 (2.80–13.90)	<0.001
CCS ≥ 6	5.36 (3.17–9.07)	<0.001
**Social History**	**Unadjusted OR** **(95% CI)**	***p*-value**
Alcohol > 10 units/week	0.44 (0.21–0.90)	0.026
**Osteoporosis**	**Unadjusted OR** **(95% CI)**	***p*-value**
Frequent Falls	2.17 (1.36–3.48)	0.001
Pre-injury Fragility Fractures	1.78 (1.07–2.97)	0.027
Post-injury Fragility Fractures	0.26 (0.10–0.67)	0.005
Osteoporosis	1.72 (1.04–2.83)	0.035
**Complications**	**Unadjusted OR** **(95% CI)**	***p*-value**
Nail Related Complications *	0.27 (0.12–0.65)	0.003
Non-union	0.23 (0.08–0.64)	0.005
HAP/CAP	3.66 (2.21–6.07)	<0.001
Post-Operative Transfusion < 48 h	2.19 (1.35–3.54)	0.001
Post-Operative Transfusion (All)	1.78 (1.07–2.98)	0.027

ISS: Injury Severity Score; CKD: Chronic kidney disease; CCS: Charlson comorbidity score; HAP: Hospital acquired pneumonia; CAP: Community acquired pneumonia. * Nail Related Complications: Included nail failure (mechanical failure at the lag screw junction, screw cut-out of the femoral head and auto-dynamisation), peri-implant fracture and peri-implant infection.

**Table 2 jcm-10-00540-t002:** Coefficients and odds ratio estimates of factors contributing to one-year mortality.

	OR	Confidence Interval	*p*-Value
Albumin (Low)	4.82	2.08–11.19	*p* < 0.001
Dementia	3.99	2.27–7.01	*p* < 0.001
HAP/CAP	3.18	1.76–5.77	*p* < 0.001
CCS > 6	2.94	1.62–5.35	*p* < 0.001

OR: Odds ratio; HAP: Hospital acquired pneumonia; CAP: Community acquired pneumonia; CCS: Charlson comorbidity score.

**Table 3 jcm-10-00540-t003:** Coefficients and odds ratio estimates outlining the risks/associations of HAP.

	OR	Confidence Interval	*p*-Value
CCS 0 to 8	1.69	1.00–2.84	*p* = 0.048
CCS ≥ 9	2.02	1.03–3.95	*p* = 0.040
Asthma/COPD	2.29	1.37–3.82	*p* = 0.002
ICU/HDU Stay	3.25	1.77–5.96	*p* < 0.001
LOS ≥ 21 days	8.82	1.18–65.80	*p* = 0.034

OR: Odds ratio; CCS: Charlson comorbidity score; COPD: Chronic obstructive pulmonary disease; ICU: Intensive care unit; HDU: High dependency unit; LOS: Length of stay.

**Table 4 jcm-10-00540-t004:** Coefficients and odds ratio estimates outlining the risks/associations of post-operative delirium.

	OR	Confidence Interval	*p*-Value
Dementia	3.84	1.98–7.44	*p* < 0.001
HAP/CAP	2.21	1.09–4.47	*p* = 0.028
UTI	3.29	1.66–6.52	*p* = 0.001
CKD Stage post-operatively (Moderate/Severe)	2.02	1.07–3.81	*p* = 0.030
ICU/HDU Stay	4.25	1.78–10.14	*p* = 0.001

OR: Odds ratio; HAP: Hospital acquired pneumonia; CAP: Community acquired pneumonia; UTI: Urinary tract infection; ICU: Intensive care unit; HDU: High dependency unit; CKD: Chronic kidney disease.

## Data Availability

Data sharing not applicable. Access to the source dataset is only permitted to employees of the Orthopaedic Department of Leeds Teaching Hospitals NHS Trush.

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
