# Peer review of "Mortality and Medical Complications of Subtrochanteric Fracture Fixation"

_jcm, 2021, doi:10.3390/jcm10030540_

Round 1

Reviewer 1 Report

The aim of this study was to report the incidence of mortality and complications in patients affected by subtrochanteric fractures, trying to identify significant risk factors related to each event. The data of 545 patients treated over an 8-year period (2009-2016) were retrospectively reviewed and many variables were included for statistical analysis.

Treatment of subtrochanteric fractures might be challenging owing to peculiar mechanical and biological problems, that increase the risk of fixation failures. Intramedullary nailing is the gold standard and was performed in all the patients of this study.

The authors should be commended for providing such a huge amount of data, but there are some major criticisms that should be highlighted.

These are my specific comments:

Title

The title does not reflect the actual contents of the paper and should be changed, because the paper investigates many conditions influencing the outcome of subtrochanteric fractures and not only the relationship between “Ageing” and “Mortality”. The authors should choose a title more congruent to the analysis performed and this would be necessarily more generic, like “Mortality and complications after subtrochanteric fractures: a retrospective analysis.” Risk assessment could be reported in the subtitle to describe and qualify the research more fully.

Abstract

The abstract should be reviewed according to the subsequent comments.

Introduction

This section is too long and should be more concise. The authors should only mention the problem of the increasing elderly population and focus the attention on the peculiar features and problems of subtrochanteric fractures, and explain why outcomes could be different from other hip fractures.

(page 5 - L 104): remove “ and wound related ”  (it’s not a systemic complication)

(page 5 - L 105): correct “ infractions “ with “ infarctions

Patients and methods

(page 5 - L 127): please provide details of institutional board approval (protocol number)

(page 5 – L132): pathological fractures were considered eligible for the analysis. In my opinion, they are a confounding factor and should be excluded from the analysis, because a pathological fracture is an independent  (but also the most important) risk factor for mortality. Moreover, the distribution of pathological fractures among the different age groups is not reported.  

(page 6 – L150-154): follow up duration is not clear. Were all the patients followed up for at least one year? How was patients’ death verified? Please specify.

Results

(page 7 - L 176): was the clinical history of all the 545 patients reconstructed? no data lacking?

(page 7 - L 177-178): the number of patients >65 years old is 411 or 434 (as written below in L 209)?

(page 7 - L 181-197): there are several incongruencies between the data reported in the text and those reported in Table 1 (eg. n. of patients with malignancy 137 vs 133, dementia 125 vs 123, smokers 113 vs 111, etc.)  Also the total number of patients is discrepant: 545 in the text, 549 in the table. Please check data accuracy and correct.

TABLE 1 should be checked and revised. Besides the overmentioned incongruencies, there are several data that are difficult to understand (eg. what is the meaning of a p-value < 0.001 for all mechanisms of injury and ASA classes?)  There are many abbreviations and some of them are not easily comprehensible: please add legenda. Why mortality rates associated with different CCS were not reported in the table? In general, Table 1 is too big and contains too much data: it should be reduced and simplified. Maybe only significant variables should be included and all the others listed in another table.

Discussion

Similarly to what said for the Introduction, the Discussion is too long and there are many parts that should be omitted, because they reaffirm well known concepts about hip fractures in elderly patients. The authors should be more concise, trying to focus specific issues related to subtrochanteric fractures or otherwise assume that there are no relevant differences with other proximal femoral fractures.

The authors should also highlight the limits of the study: it is retrospective and it is difficult to evaluate the specific role of the various factors influencing the prognosis of these patients. Moreover, some relevant data are lacking, like the type of anaesthesia, BMI or type of VTE prophylaxis.

The paper should preferably represent a descriptive analysis and any inferential consideration should be made cautiously.

(page 14 - L 326-329): it’s probably too optimistic to assert that it is possible to treat or prevent risk factors in these patients. It would be more realistic to say that these data might provide some elements of prognostic value for these patients.

In conclusion, the paper needs major revisions to be accepted for publication. The authors should review the manuscript in order to improve the scientific relevance of the collected data.  Pathological fractures should be excluded from the analysis. Any incongruence in the numbers reported in the text and in the tables must be identified and corrected.

Introduction and Discussion should be shortened and centered on specific problems related to subtrochanteric fractures, emphasizing potential differences with other hip fractures.

Reviewer 2 Report

Review for Journal of Clinical Medicine

Title: The impact of ageing on mortality following subtrochanteric fractures

The authors present o retrospective analysis of a patient cohort with subtrochanteric fractures. In general this is an interesting topic, because subtrochanteric fractures are challenging injuries and complications are not infrequent.

General notes:

  • The manuscript could perhaps be shortened, especially the introduction and discussion sections.
  • All abbreviations should be explained at first use and a list with all used abbreviations should be added to the manuscript.
  • The time from admission to surgery should be analysed as a risk factor for mortality and complications as well in my opinion.
  • The manuscript contain some unusual terms and grammar (e. g. “commonest, line 103 or line 85-86, line 131-132, line 397-399). Critical revision of speech and grammar is recommended.
  • There are several discrepancies in the results section (see specific notes), which should be explained or corrected before publication.

Specific notes:

  • Line 44: “chest infection” = better use “pneumonia”?
  • Line 47-48: “within this group” -> which group?
  • Line 65-77 could be shortened. The demographic changes are widely known.
  • Line 78-79: This is not correct I think, most common injuries are distal radius fractures.
  • Line 105: “infraction” -> infarction
  • Like mentioned above I miss the risk factor “time from admission to surgery” in this manuscript.
  • Methods: How was it possible for you to analyse 1-year mortality? Where do you have the information about the patients, especially of those who had no follow-up of more than 1 year? Do you excluded all patients with follow-up under 1 year (which is not mentioned in exclusion criteria)?
  • Line 165: “Kaplan-Meier”
  • Line 186 – 188: The sum of the mentioned patient groups is < 545, do you have information about the home situation of the other patients?
  • Line 176, 178 and line 209 and table 1: There are different results for the same parameter (patients > 65 years): 411 versus 434 patients. In table 1 the mentioned group is 415 in sum.
    Also line 176 (total 545 patients) and sum of table 1 (549 patients) is divergent. This is not plausible. I haven’t checked every single number of table 1, so perhaps more mistakes can be found.
  • Line 225: A definition of “nail complications” should be given.
  • Line 242 and 249: “22” and “twenty-three”. Consistent use of numbers recommended.
  • Table 1: Definition of “anaesthetic time”?
  • Line 283-284: I think this is biased, because the incidence of “multiple injuries” should be higher in young patients.
  • Line 347: 2x “in” -> of
  • Line 432: double comma
  • Line 489: Is “(9)” referring to reference no. [9]? I think this could be a mix-up.
  • Line 476-496: How delirium was assessed in this patient cohort? Hypoactive delirium is difficult to recognize and can be a risk factor for poor outcome.
  • Line 512: delete “other”
  • Line 534: “accessed on …”
  • Research of the last two years should be integrated in the introduction and discussion sections.

Example: Risk factors for nonunion after intramedullary nailing of subtrochanteric femoral fractures.

Krappinger D, Wolf B, Dammerer D, Thaler M, Schwendinger P, Lindtner RA.

Arch Orthop Trauma Surg. 2019 Jun;139(6):769-777. doi: 10.1007/s00402-019-03131-9. Epub 2019 Feb 7. PMID: 30729990

Round 2

Reviewer 1 Report

In my opinion, the revised version of the paper is suitable for publication in the present form.

However, I don't understand why there are four Kaplan-Meier survival curves in Figure 1. There are two curves for age and two curves for gender. I read different numbers of patients for each parameter, but I don't understand what they represent. Please explain or correct. 

Author Response

We thank the reviewer for their comment. There should only be only one Kaplan-Meier survival curve for gender and one for age (Figure 1). We have "accepted" all tracked changes on the manuscript to avoid confusion. 

Reviewer 2 Report

Two curves in Figure 1 contain the "old data before revision" and should be removed.

All major problems of the manuscript have been adressed and it could be accepted in the present form.

For easier reading I would additionally favour a manuscript without tracked changes.

Author Response

We thank the reviewer for their comment. There should only be only one Kaplan-Meier survival curve for gender and one for age (Figure 1). We have "accepted" all tracked changes on the manuscript to avoid confusion, as per reviewer's suggestion.